# The Design of the Patellar Component Does Not Affect the Patient-Reported Outcome Measures in Primary Posterior-Stabilized Total Knee Arthroplasty: A Randomized Prospective Study

**DOI:** 10.3390/jcm11051363

**Published:** 2022-03-02

**Authors:** Oog-Jin Shon, Gi Beom Kim

**Affiliations:** 1Department of Orthopedic Surgery, Yeungnam University College of Medicine, 170 Hyonchung-ro, Namgu, Daegu 42415, Korea; maestro-jin@hanmail.net; 2Department of Orthopedic Surgery, Yeungnam University Medical Center, 170 Hyonchung-ro, Namgu, Daegu 42415, Korea

**Keywords:** knee, total knee arthroplasty, patellar resurfacing, dome type, anatomic type

## Abstract

This randomized comparative study was conducted to investigate the outcomes of patellar resurfacing with a medialized dome or an anatomical type in patients receiving primary unilateral posterior-stabilized TKA. Between March 2019 and January 2021, 98 knees were randomly assigned to receive patellar resurfacing by a medialized dome type (group D, 49 knees) or an anatomic type (group A, 49 knees). The primary outcome was the Knee Injury and Osteoarthritis Outcome Score. The secondary outcomes were the Western Ontario and McMaster Universities Osteoarthritis Index, Feller’s patella score, the Kujala anterior knee pain score, knee joint range of motion (ROM), and postoperative complications, including periprosthetic patellar fracture, patellar tilt angle, and lateral patellar shift. Patient-reported outcomes were not significantly different between the two groups. The ROM of the knee joint was significantly better in group A at six months after surgery (*p* = 0.021). No complications such as patellar fractures were observed. The anatomic type of patellar component showed a significant improvement of the patellar tilt angle after surgery compared with the medialized dome type of component. However, there were no significant differences in patient-reported clinical outcomes between the two groups during the follow-up period of 12 months.

## 1. Introduction

Various designs of patellar components have been used in total knee arthroplasty (TKA), including dome, medialized dome, oval, and anatomical types [1,2,3]. They vary from implant to implant or even within the same implant, and most are made of ultra-high molecular weight polyethylene [4,5]. To our knowledge, medialized dome-type components have been most commonly used because of their ease of operation due to the absence of an axis of rotation for patellar tracking [1,4].

With the recent development of TKA design, an anatomic type of patellar component, with the theoretical advantages of optimizing the conformity of the patello-femoral (P-F) joint, promoting native patellar tracking, and improving quadriceps function, has been developed [6,7,8,9]. It has been suggested that P-F symptoms can be reduced by optimizing kinematics [2,5,10].

To our knowledge, some studies related to P-F kinematics according to patellar component design have been reported [5,7,8]. However, even if a patellar component with this advanced design has the theoretical advantages of kinematics, there has been a paucity of literature regarding whether it leads to clinically improved patient-reported outcomes.

Therefore, the purpose of this randomized comparative study was to investigate the outcomes of patellar resurfacing with a medialized dome or an anatomical type in patients receiving primary unilateral posterior-stabilized TKA. We hypothesized that, although the anatomical type of patellar component had kinematic advantages, it would not affect patient-reported outcomes.

## 2. Materials and Methods

### 2.1. Participants

This prospective comparative study was ethically approved by the institutional review board of our hospital before gathering the patient data. Informed consent was obtained from all patients for study enrollment. From March 2019 to January 2021, 103 knees that underwent unilateral primary TKA with patellar resurfacing using a medialized dome type or an anatomic type were screened. Indications for patellar resurfacing were as follows: patients with progressed patellofemoral osteoarthritis (OA) according to the Iwano classification system (stage III–IV) [11], with a resultant thickness greater than 12 mm after patellar resection, considering the native patellar thickness [12]. We enrolled patients with a 12-month follow-up after the index operation for the study. We excluded five knees for the following reasons: rheumatoid arthritis in two, previous arthroscopic surgery around the knee in two, and post-traumatic OA in one. Finally, 98 knees fully meeting the inclusion criteria were randomly assigned either to the group with a medialized dome type (group D, 49 knees) or to the group with an anatomic type (group A, 49 knees). All patients completed the 12-month follow-up (Figure 1).

### 2.2. Surgical Techniques

All operations were performed by the same surgeon (G.B.K.) in our hospital using the modified gap-balancing technique, which can balance the extension gap before the flexion gap with posterior-stabilized (PS) gradually reducing radius femoral implants (Attune^®^ TKA System, Depuy Synthes Inc., Warsaw, IN, USA) [13,14]. A medial parapatellar arthrotomy with a midline incision was performed. Femur sizing was performed using an anterior reference system in all the cases. The rotation of the tibial component was set by considering several reference points, including the medial one-third of the tibial tuberosity, anterior tibial cortex, and floating technique. All prostheses were used with cement. Fixed-bearing antioxidant polyethylene inserts were used in all cases.

The patellar components in the Attune^®^ TKA System are composed of a medialized dome-type or an anatomic-type patellar component (Figure 2) [5].

The thickness of the patella was measured using a vernier caliper before cutting in all patients. The amount of resection was decided using a cutting jig of either 7.5 mm or 9.5 mm, depending on the native patellar thickness, which corresponded to the thickness of the patellar component to be used. In the medialized dome-type components, cutting was set regardless of the axis of patellar tracking, whereas in the anatomic type, the longitudinal axis of the patella was set to be parallel to the tibial component in order to precisely match the rotational axis of the patellar tracking (Figure 3).

The size of the component was determined to provide maximum coverage of the articular surface. After resection, the resultant thickness was measured using a vernier caliper. Moreover, patelloplasty, including the removal of marginal osteophytes, and circumferential denervation using electrocautery were performed. Intraoperative patellar tracking was checked throughout the knee motion with the no thumb technique [15].

### 2.3. Postoperative Protocols

A closed suction drain was inserted and was removed 24 h after surgery. All patients performed the same perioperative pain-control protocol, including a multimodal drug regimen, postoperative patient-controlled analgesia, and an intraoperative periarticular injection. Active and passive postoperative range of motion (ROM) exercises were started on the day of surgery. If the acute pain subsided, partial weight-bearing with a crutch was allowed on the first postoperative day. Full weight-bearing was permitted 3 weeks after surgery.

### 2.4. Outcome Assessments

Demographic characteristics were investigated before surgery. All patients were regularly followed up at six weeks, and at three, six, and 12 months after index surgery. Patient-reported clinical evaluations were performed using the Knee Injury and Osteoarthritis Outcome Score (KOOS) [16], the Western Ontario and McMaster Universities Osteoarthritis Index (WOMAC) [17], Feller’s patella score [18], the Kujala anterior knee pain (AKP) score [19], the ROM of the knee joint (including flexion contracture and further flexion), and postoperative complications including periprosthetic patellar fracture. Clinical outcomes were assessed at regular follow-ups by an independent observer in an outpatient clinic.

Radiographic images were retrieved using a picture-archiving and communication system (PACS; IMPAX, Agfa Healthcare, Mortsel, Belgium), and radiographic measurements were performed at regular follow-ups by another independent observer of the operative team. Radiographic evaluations included the patellar tilt angle and the lateral patellar shift on a Merchant view radiograph. The patellar tilt angle was set as the angle between the line crossing the widest portion of the patella and the line passing through the most anterior surface of both femoral condyles [20,21].

The primary outcome was KOOS. The secondary outcomes were: (1) WOMAC; (2) Feller’s patella score; (3) Kujala AKP score; (4) knee joint ROM; (5) postoperative complications including periprosthetic patellar fracture; (6) patellar tilt angle; and (7) lateral patellar shift.

### 2.5. Statistical Analysis

A power analysis (G* power software, version 3.1.9) was performed to calculate the number of patients needed in each group to identify the significant differences in clinical outcomes. We designed the study with a power of 90% at a two-sided significant level of 7%, which indicated that 44 knees were necessary in each group. Finally, considering an anticipated dropout rate of 10%, a total of 98 knees were required.

Statistical evaluation was performed using IBM SPSS software version 23 (IBM Corp., Armonk, NY, USA), and continuous data were expressed as means with SDs. All dependent variables were tested for normality of distribution and equality of variances using the Kolmogorov–Smirnov test and analyzed using parametric or non-parametric tests based on normality. An independent samples *t*-test (parametric) and a Mann–Whitney U-test (non-parametric) were performed to assess the differences in clinical and radiographic variables between the two groups. Fisher’s exact test was used to compare the ratios between the groups. For all tests, the statistical significance was set at *p* < 0.05.

## 3. Results

This A total of 98 knees were recruited into the study, 49 of which received patellar resurfacing using a medialized dome-type patellar component (group D), and 49 of which received with an anatomic-type patellar component (group A). There was no significant difference in the baseline characteristics between the two groups (Table 1).

Patient-reported outcomes were significantly improved in both groups at 12 months after surgery (*p* < 0.001). All clinical outcomes were not significantly different between the two groups at each time period (Table 2, Table 3 and Table 4).

Further flexion of the knee joint was significantly better in group A at six months after surgery (*p* = 0.021), but not at 12 months after surgery (Table 5).

No complications such as patellar fractures were observed during the follow-up period.

The preoperative patellar tilt angle and lateral patellar shift did not differ significantly between the groups. However, postoperative patellar tilt angle was significantly improved in group A compared with group D and was consistent during the follow-up period (*p* < 0.001) (Figure 4).

## 4. Discussion

This randomized prospective comparative study showed that, although the anatomic-type patellar component showed significant radiological improvement after surgery, it did not lead to significant clinical improvements in Asian people.

Since the anatomic patellar component has a highly conforming geometry in the trochlear of the femoral component by implementing more anatomic patellar bone with a medialized apex, sagittal plane kinematics and quadriceps performance can be improved by optimizing native PFJ tracking. These factors were also reflected in the radiological outcomes of this study, suggesting that the postoperative patellar tilt angle was significantly improved. A study that revealed differences in patellar kinematics according to patellar component design reported that patellar tilt was reduced in the anatomic-type patellar component for each angle of the knee flexion [7]. This can be attributed to the fact that the dome-type component has a larger medial and lateral contact area compared with that of the anatomic type [9]. However, despite these kinematic advantages, since anatomic-type patellar components are usually more sensitive to mal-positioning such as rotational error [4], effort is needed to accurately align the tracking axis (Figure 3). If the central ridge of the anatomical-type component is not rotationally aligned with the femoral trochlea, adverse effects such as polyethylene wear or increased force in the extensor mechanism may occur. On the other hand, in the case of the medialized dome type, it can be implemented regardless of the rotational axis for patellar tracking.

Some studies related to the kinematic effects of different patellar component design reported that the anatomic type of patellar component had greater PF joint flexion angle with reduced contact force during activities [7,8,9,22]. This was also demonstrated in the results of this study, in which knee ROM was significantly improved in group A up to 6 months after surgery. However, there was no statistically significant difference in knee ROM at 12 months after surgery, which seems to have reduced the difference between the two groups as the pain subsides, the swelling of the surrounding tissue decreases, and the muscle power improves.

Moreover, the patient-reported clinical outcomes of the current study did not differ between the two groups. Theoretical kinematic advantages due to subtle changes in patellar component design did not lead to a significant improvement in short-term clinical outcomes. In the future, it is necessary to evaluate patient-reported outcomes over a longer term.

Despite the informative results, there are limitations meriting discussion. First, this study had a relatively short-term follow-up. Accordingly, significant differences may have been missed, and we could not confirm that the results of this study guarantee mid- to long-term outcomes. Another limitation of this study was the small sample size. The sample size of this study was determined by the effect size based on the KOOS score of previous similar studies. Therefore, a randomized controlled trial with a large sample and longer-term follow-up is needed. Third, since this study was conducted with only a specific type of implant (fixed-bearing PS implant), the results of this study cannot be representative of all TKA implants. The outcomes may vary for other implants or manufacturers with different kinematics. Lastly, a female predominance was observed in this study. Thus, the same outcomes may not apply to populations with different sex ratios. This trend has been characteristic of the Asian population [23,24].

Nevertheless, the strengths of this study are as follows: first, to our knowledge, this is the first prospective study to report the outcomes of a medialized dome-type and an anatomic-type patellar component in Asian people; secondly, patients were randomly assigned, and a variety of patient-reported clinical assessments were performed. Therefore, the patient-reported outcomes can be considered sufficiently reliable.

## 5. Conclusions

In conclusion, the anatomic type of patellar component showed a significant improvement of the patellar tilt angle after surgery compared with the medialized dome type of component. However, there were no significant differences in patient-reported clinical outcomes between the two groups during short-term follow-up.

## Figures and Tables

**Figure 1 jcm-11-01363-f001:**
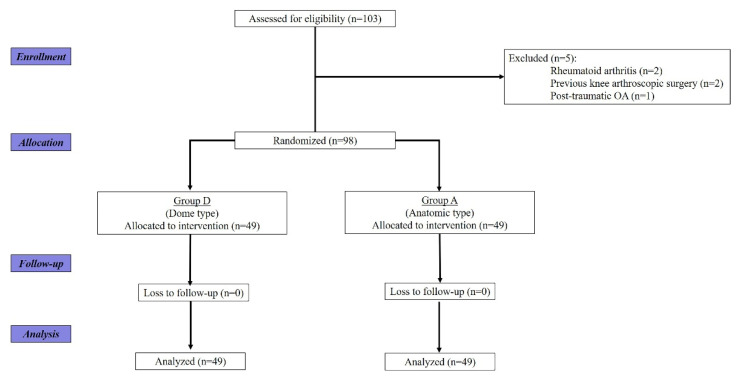
Consolidated Standards of Reporting Trials (CONSORT) flow diagram for enrolled patients.

**Figure 2 jcm-11-01363-f002:**
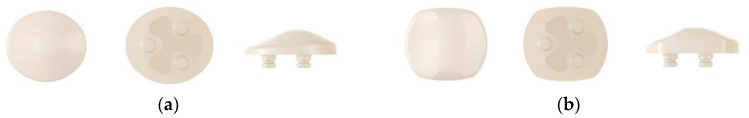
Anteroposterior and lateral views of two types of patellar components. (**a**) Medialized dome type and (**b**) anatomic type.

**Figure 3 jcm-11-01363-f003:**
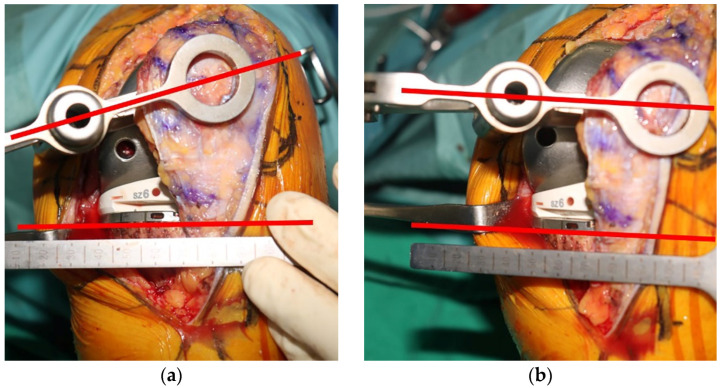
Determining method of the tracking axis of anatomic-type patellar components. (**a**) If the longitudinal axis of the patella is not parallel to the tibial component, the tracking axis is considered inappropriate, and (**b**) if it is parallel, it is considered appropriate.

**Figure 4 jcm-11-01363-f004:**
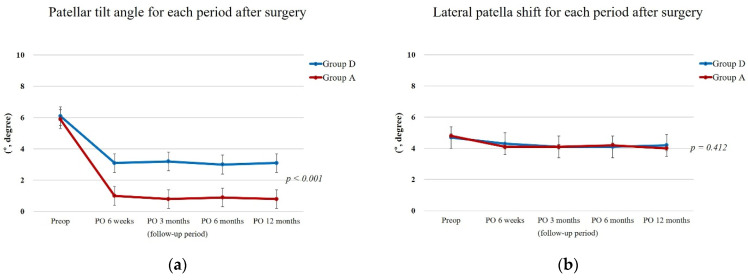
Radiographic outcomes for each period after surgery. (**a**) Patellar tilt angle. (**b**) Lateral patella shift. Postoperative patellar tilt angle was significantly improved in group A compared with group D. (An independent sample *t*-test was used to assess the differences in clinical and radiographic variables between the two groups. *p* value < 0.05 was considered statistically significant).

**Table 1 jcm-11-01363-t001:** Baseline patient characteristics for the medialized dome type (group D) and anatomic type (group A).

Variables	Total	Group D	Group A	*p* Value
Age, years ^1^	71.6 (60–85)	71.8 (60–85)	71.3 (61–82)	0.671
Sex, *n* ^2^				
Female, *n*	88 (89.8)	43 (87.8)	45 (91.8)	0.505
Male, *n*	10 (10.2)	6 (12.2)	4 (8.2)	
BMI, kg/m^2 1^	27.9 (19.2–33.4)	27.6 (19.2–30.5)	28.1 (20.1–33.4)	0.817
F/U period, months	12.0	12.0	12.0	-
Side, *n* ^2^				
Right, *n*	46 (46.9)	22 (44.9)	24 (49.0)	0.686
Left, *n*	52 (53.1)	27 (55.1)	25 (51.0)	
Preop HKA angle, (°) ^1^	−5.6 (−21.0–13.5)	−5.7 (−21.0–12.5)	−5.5 (−16.0–13.5)	0.591
Iwano classification, *n* ^2^				
Stage III	69 (70.4)	35 (71.4)	34 (69.4)	0.825
Stage IV	29 (29.6)	14 (28.6)	15 (30.6)	

Note: ^1^ Data are presented as means (range). ^2^ Data are presented as numbers (percentage). A negative value of HKA angle indicated varus alignment. The level of statistical significance was set at *p* < 0.05. BMI: body mass index; F/U: follow-up; Preop: preoperative; HKA: hip-knee-ankle.

**Table 2 jcm-11-01363-t002:** Mean Knee Injury and Osteoarthritis Outcome Score (KOOS) for group with the medialized dome type (group D) and group with anatomic type (group A).

Variables	Group D (*n* = 49)	Group A (*n* = 49)	*p* Value ^1^
**KOOS Pain**			
Preop	49.2 ± 6.4	48.9 ± 7.5	0.631
PO at 3 months	80.7 ± 8.7	81.0 ± 8.0	0.498
PO at 6 months	85.8 ± 6.5	86.1 ± 6.1	0.702
PO at 12 months	86.4 ± 4.9	86.2 ± 5.3	0.681
**KOOS Symptom**			
Preop	46.2 ± 8.8	47.1 ± 6.5	0.249
PO at 3 months	60.2 ± 5.1	59.8 ± 5.3	0.392
PO at 6 months	60.8 ± 4.5	60.4 ± 4.9	0.681
PO at 12 months	61.2 ± 3.2	61.0 ± 4.7	0.783
**KOOS QOL**			
Preop	26.7 ± 9.2	26.9 ± 7.5	0.813
PO at 3 months	68.5 ± 7.6	69.1 ± 4.4	0.241
PO at 6 months	70.2 ± 4.1	70.1 ± 4.7	0.671
PO at 12 months	70.6 ± 4.6	70.5 ± 4.5	0.712
**KOOS Function**			
Preop	52.6 ± 8.2	53.1 ± 7.7	0.245
PO at 3 months	81.6 ± 9.2	82.2 ± 8.5	0.104
PO at 6 months	81.8 ± 6.4	83.4 ± 6.2	0.093
PO at 12 months	85.6 ± 6.5	85.9 ± 7.0	0.439

Note: Data are presented as means ± standard deviation. The level of statistical significance was set at *p* < 0.05. ^1^ As analyzed with the independent samples *t*-test, there were no significant differences between the groups. Preop: preoperative; PO: postoperative; QOL: quality of life.

**Table 3 jcm-11-01363-t003:** Mean Western Ontario and McMaster Universities Osteoarthritis Index (WOMAC) for group with the medialized dome type (group D) and group with anatomic type (group A).

Variables	Group D (*n* = 49)	Group A (*n* = 49)	*p* Value ^1^
**WOMAC Function**			
Preop	46.6 ± 8.1	47.1 ± 7.2	0.695
PO at 3 months	70.9 ± 9.9	73.2 ± 8.5	0.130
PO at 6 months	77.9 ± 7.5	81.5 ± 8.1	0.182
PO at 12 months	86.9 ± 6.9	87.3 ± 7.5	0.757
**WOMAC Pain**			
Preop	46.4 ± 6.2	47.2 ± 6.4	0.359
PO at 3 months	76.5 ± 9.7	76.3 ± 8.0	0.812
PO at 6 months	84.5 ± 6.1	84.1 ± 6.5	0.790
PO at 12 months	88.7 ± 4.9	89.5 ± 4.3	0.673
**WOMAC Stiffness**			
Preop	40.2 ± 8.4	40.9 ± 9.2	0.635
PO at 3 months	61.7 ± 7.1	62.2 ± 7.5	0.684
PO at 6 months	68.8 ± 8.0	69.1 ± 7.4	0.728
PO at 12 months	81.2 ± 6.9	81.3 ± 6.2	0.898

Note: Data are presented as means ± standard deviation. The level of statistical significance was set at *p* < 0.05. ^1^ As analyzed with the independent samples *t*-test, there were no significant differences between the groups. Preop: preoperative; PO: postoperative.

**Table 4 jcm-11-01363-t004:** Mean Feller’s patella score and mean Kujala anterior knee pain score for group with the medialized dome type (group D) and group with anatomic type (group A).

(a) Feller’s Patella Score			
Variables	Group D (*n* = 49)	Group A (*n* = 49)	*p* Value ^1^
Preop	11.7 ± 10.3	12.1 ± 9.2	0.714
PO at 3 months	22.4 ± 6.5	23.8 ± 7.2	0.203
PO at 6 months	24.0 ± 4.9	24.5 ± 5.6	0.382
PO at 12 months	27.4 ± 5.3	27.5 ± 4.2	0.496
(b) Kujala Anterior Knee Pain Score			
Variables	Group D (*n* = 49)	Group A (*n* = 49)	*p* Value ^1^
Preop	68.4 ± 9.8	68.6 ± 10.3	0.762
PO at 3 months	71.9 ± 8.2	72.8 ± 7.7	0.446
PO at 6 months	76.1 ± 6.2	78.5 ± 7.2	0.230
PO at 12 months	80.1 ± 4.2	79.9 ± 5.1	0.418

Note: Data are presented as means ± standard deviation. The level of statistical significance was set at *p* < 0.05. ^1^ As analyzed with the independent samples *t*-test, there were no significant differences between the groups. Preop: preoperative; PO: postoperative.

**Table 5 jcm-11-01363-t005:** Mean ROM of the knee joint for group with the medialized dome type (group D) and group with anatomic type (group A).

Variables	Group D (*n* = 49)	Group A (*n* = 49)	*p* Value
**FC** (°)			
Preop	9.2 ± 3.2	10.1 ± 3.0	0.317
PO at 3 months	1.7 ± 4.2	1.8 ± 4.5	0.623
PO at 6 months	1.9 ± 1.8	1.7 ± 2.1	0.576
PO at 12 months	1.5 ± 2.7	1.6 ± 2.2	0.896
**FF** (°)			
Preop	118.7 ± 4.5	119.2 ± 5.3	0.735
PO at 3 months	125.8 ± 2.8	130.7 ± 2.7	0.513
PO at 6 months	126.3 ± 3.3	136.9 ± 2.5	**0.021** ^1^
PO at 12 months	135.6 ± 2.1	136.8 ± 1.9	0.715

Note: Data are presented as means ± standard deviation. The level of statistical significance was set at *p* < 0.05. ^1^ As analyzed with the independent samples *t*-test, FFof the knee joint was significantly better in group A at six months after surgery. The maximum value of FF was set at 140 degrees. ROM, range of motion; FC, flexion contracture; FF, further flexion; Preop, preoperative; PO, postoperative.

## Data Availability

Data supporting the reported findings are available from the corresponding author upon reasonable request.

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
