# Peer review of "The Design of the Patellar Component Does Not Affect the Patient-Reported Outcome Measures in Primary Posterior-Stabilized Total Knee Arthroplasty: A Randomized Prospective Study"

_jcm, 2022, doi:10.3390/jcm11051363_

Round 1
Reviewer 1 Report
General Comments: The authors present a randomized study comparing the outcomes of medialized dome vs anatomic patellar components in patients undergoing unilateral total knee arthroplasty with posterior-stabilized implants. Despite a significant improvement in patellar tilt angle with the anatomic group, they found no significant difference in functional outcomes or complications at 12 months between the two groups.
Overall, a well-designed, executed, and written study. The authors should be commended for their efforts.
Specific Comments:
Abstract:
Concise yet fully explains the purpose of the study and significant findings. Would recommend listing the follow-up period as 12 months opposed to just stating 'short-term follow-up'
Line 12: Remove first sentence "A single paragraph..."
Intro:
Sufficiently presents the topic and purpose of study
Methods:
Line 61: '120 knees' should be '98 knees'
Results:
Clearly presents data
Table 1: Correct horizontal header to state 'Group D', 'Group A', and 'Total' instead of Title 2, Title 3, and Title 4
Discussion:
Adequately discusses findings and their impact
Author Response
Reviewer #1:
General Comments: The authors present a randomized study comparing the outcomes of medialized dome vs anatomic patellar components in patients undergoing unilateral total knee arthroplasty with posterior-stabilized implants. Despite a significant improvement in patellar tilt angle with the anatomic group, they found no significant difference in functional outcomes or complications at 12 months between the two groups.
Overall, a well-designed, executed, and written study. The authors should be commended for their efforts.
Response: Thank you for your positive comments.
Specific Comments:
Abstract:
Concise yet fully explains the purpose of the study and significant findings. Would recommend listing the follow-up period as 12 months opposed to just stating 'short-term follow-up'
Response: Following your comments, we have revised the conclusion as “the follow-up period as 12 months”. Thank you.
Line 23-25: However, there were no significant differences in patient-reported clinical outcomes between the two groups during the follow-up period of 12 months.
Line 12: Remove first sentence "A single paragraph..."
Response: Following your comments, we have deleted the sentence. Thank you.
Intro:
Sufficiently presents the topic and purpose of study
Response: Thank you for your positive comments.
Methods:
Line 61: '120 knees' should be '98 knees'
Response: Following your comments, we have revised the sentence. Thank you.
Line 61: Finally, 98 knees fully meeting the inclusion criteria…
Results:
Clearly presents data
Response: Thank you for your positive comments.
Table 1: Correct horizontal header to state 'Group D', 'Group A', and 'Total' instead of Title 2, Title 3, and Title 4
Response: Following your comments, we have revised the table 1. Thank you.
Discussion:
Adequately discusses findings and their impact
Response: Thank you for your positive comments.
Reviewer 2 Report
I commend the authors for performing this interesting research.
Generally, the manuscript is well written and easy to follow. However, the follow-up period is very short (at least 2-year FU is expected for serious studies on arthroplasty).
My suggestions to further improve/clarify the manuscript are the following:
(1) Is patellar resurfacing always necessary or were all TKAs performed with patellar resurfacing just for the purpose of this study?
(2) There are considerable differences in patellar tracking in varus and valgus knees. Were all included knees of varus malalignment?
(3) Anterior knee pain (and patella baja) were attributed to the handling of infrapatellar fat pad. How did you handle this during the surgery?
Technically, Table 1 is inappropriate since Titles 2-4 are not specified.
Graphically, a photograph of anatomical- and dome-type patella button would be much appreciated.
Author Response
Reviewer #2:
I commend the authors for performing this interesting research.
Response: Thank you for your positive comments.
Generally, the manuscript is well written and easy to follow. However, the follow-up period is very short (at least 2-year FU is expected for serious studies on arthroplasty).
Response: Thank you for your positive comments. We are also fully aware of the short follow-up period, which is a limitation of this study. In the future we will report on its mid-to long term outcomes. thank you.
My suggestions to further improve/clarify the manuscript are the following:
- Is patellar resurfacing always necessary or were all TKAs performed with patellar resurfacing just for the purpose of this study?
Response: There seems to be a misunderstanding about this content. As we had described in the method section of the article, only in patients with advanced patellofemoral osteoarthritis [Iwano classifica-tion system (stage III‒IV)] and the resultant thickness greater than 12 mm after resection considering the native patellar thickness, we had performed patellar resurfacing. Therefore, patellar resurfacing was not always necessary.
(2) There are considerable differences in patellar tracking in varus and valgus knees. Were all included knees of varus malalignment?
Response: We have fully agreed with your comments. As you comment, the patellar tracking is quite different in varus and valgus knees. We included both varus and valgus knees in the present study. We have added the preoperative HKA angle to table 1. Thank you.
- Anterior knee pain (and patella baja) were attributed to the handling of infrapatellar fat pad. How did you handle this during the surgery?
Response: We have fully agreed with your opinion. We had removed some of the infrapatellar fat pad during surgery if it was judged that it caused impingement or irritation to the implant. Nevertheless, we do not think it is appropriate to describe this in a surgical technique for the conciseness of the article.
Technically, Table 1 is inappropriate since Titles 2-4 are not specified.
Response: This was our mistake and we are very sorry. Following your comments, we have revised the table 1. Thank you.
Graphically, a photograph of anatomical- and dome-type patella button would be much appreciated.
Response: Following your comments, we have added the picture of two types of patellar components. In particular, we have obtained permission from Knee Joint Part of Johnson & Johnson, DePuy Synthes, for use of this figure. Thank you.